# Effect of 6-Gingerol on Oxidation and Structure of Beef Myofibrillar Protein During Heating

**DOI:** 10.3390/foods14071081

**Published:** 2025-03-21

**Authors:** Ruhong Bai, Anguo Xie, Han Wu, Kun Zhang, Shubei Dong, Yunhong Liu

**Affiliations:** 1College of Food and Bioengineering, Henan University of Science and Technology, Luoyang 471023, China; bairuhong0508@163.com (R.B.); anguo@nyist.edu.cn (A.X.); 19837950935@163.com (S.D.); 2Zhang Zhongjing School of Chinese Medicine, Nanyang Institute of Technology, Nanyang 473003, China; hanwu0808@gmail.com; 3Nanyang Biaodian Food Co., Ltd., Nanyang 474250, China; kunzhang353276@sina.com

**Keywords:** beef, 6-gingerol, oxidation, myofibrillar protein, temperature, protein structure

## Abstract

High-temperature cooking can induce oxidation and structural changes in myofibrillar protein (MP), harming meat product quality. 6-gingerol is a key part of ginger and a natural antioxidant. In this study, MP was mixed with 6-gingerol and cooked at different temperatures. Chemical methods, fluorescence spectroscopy, Fourier transform infrared spectroscopy (FT-IR), scanning electron microscopy (SEM), and molecular docking were used to study the effects on protein aggregation, oxidation, molecular structure, and the microstructure of muscle fibers. The results showed that 40 μg/mL of 6-gingerol significantly optimized the indexes of beef MP. For example, 6-gingerol inhibited the decrease in MP sulfhydryl content and solubility, delayed the rise in surface hydrophobicity and carbonyl content, decreased the particle size of MP, and elevated the absolute value of Zeta potential, which, in turn, hindered oxidative denaturation and the aggregation of proteins. 6-gingerol could maintain the stability of the spatial conformational structure and microstructure of the protein. The protein secondary structure changed, and the α-helical might have been transformed into the β-folded one. The binding of 6-gingerol to MP mainly relied on hydrogen bonds, van der Waals forces, and hydrophobic interactions. Thus, 6-gingerol had a positive effect on the antioxidant properties and structural stability of beef MP during heating.

## 1. Introduction

Beef, which is regarded as an excellent source of protein, enjoys great popularity among consumers all over the world due to its abundant nutritional value. *Longissimus thoracis* (*LT*) is one of the most dominant muscles in beef, extremely high in protein, widely available, and very cost effective. Myofibrillar protein (MP), the most abundant in beef, is an important component of myofibrils and accounts for a significant proportion of total beef protein. It is a complex mixture including myosin and actin, essential for muscle fiber integrity and functionality. MP also plays a crucial role in the rheological properties, water retention, emulsification, texture, sensory, and other qualities of meat products [1]. Heat treatment is a common method for meat products before their consumption. It endows meat products with a charming aroma and pleasant taste. However, improper cooking may also have some adverse effects on food. The heating of beef during cooking may lead to the oxidization of MP and change its structure. This change will unfold the molecular structure and expose the amino acid side-chain groups, which will trigger a series of reactions [2], such as the generation of carbonyl derivatives and disulfide bonds, which, in turn, will cause changes in their physicochemical and functional properties and seriously affect the quality of meat products in terms of taste, color, odor, and texture. Heating also induces the thermal aggregation of proteins, making their structures less likely to be recognized and degraded by digestive enzymes, leading to a reduced digestibility and bioavailability [3]. It has also been shown that the natural nutrients hidden inside meat products may be lost after heating [4], which leads to the production of pentanes and aldehydes (including hexanal, malondialdehyde, and 4-hydroxynonenal). These small molecules may react with the ε-amino group of proteins in a Melad reaction, resulting in changes in the color, textural properties, and flavor of meat products, which may reduce their nutritional value or even endanger human health [5]. Wang et al. [6] found that the oxidative cross-linking that occurs in proteins causes changes such as muscle contraction, which alters water distribution and migration, which, in turn, affects meat tenderness. Pang et al. [7] found that the denaturation of myofibrillar proteins plays a key role in cooking losses, which means that there is a strong correlation between the heating temperature of the meat and the denaturation of proteins.

In order to reduce the adverse effects of protein oxidation during heating, antioxidants are often used to control the course of oxidation reactions in meat products. Antioxidants can prevent harmful substances from spoiling, reduce nutrient loss while maintaining a better appearance, and prevent unpleasant odors from oxidized proteins. Due to food safety issues, synthetic antioxidants such as butylated hydroxyanisole (BHA) and dibutylated hydroxytoluene (BHT) have good antioxidant effects and are inexpensive, but have potential cytotoxicity and carcinogenicity risks with long-term consumption [8]. As a result, green and safe natural antioxidants with a good antioxidant capacity are being increasingly used in the food industry.

Ginger (*Zingiber officinale* Roscoe) is a herbaceous perennial plant of the genus *Zingiber officinalis* in the family Zingiberaceae, which is used for both medicine and food [9]. As a common spice in meat processing, ginger not only plays the roles of deodorizing, flavoring, improving tenderness, and enhancing water retention in beef cooking, but also inhibits a variety of unfavorable chemical reactions, such as lipid oxidation and acrylamide production [10]. 6-gingerol has the highest relative content of phenolic compounds in curcumin components, accounting for up to 24.42%, is a natural antioxidant, and is safe and non-toxic, with the advantages of low extraction costs and extraction process maturity, in line with consumer demand for natural food additives [11]. 6-gingerol is a phenolic hydroxyl-containing substance with weak polarity, which is a functional factor of dried ginger’s variety of pharmacological effects [12], with antioxidant, anti-inflammatory, anti-tumor, hypolipidemic, and bactericidal and antiseptic effects. It was shown that 6-gingerol paired with perilla oil significantly inhibited lipid and protein oxidation and the microbial growth of surimi during refrigerated storage [13]. Li et al. [14] found that the addition of moderate amounts of 6-gingerol improved the texture, water-holding capacity, and microstructure of lamb meatballs. In recent years, there have been numerous studies employing natural antioxidants to mitigate the oxidation of meat products in thermal processing, such as tea polyphenols (with strong in vitro antioxidant activity), which can improve protein oxidation and fat oxidation in grilled beef patties and grilled pork patties, thus improving their quality [15,16]. Bamboo leaf extracts were similarly effective in inhibiting protein and fat oxidation in thermally processed meat products, while bamboo leaf extracts themselves had weak antioxidant activity in vitro [17]. Thus, there is not necessarily a significant correlation between the in vitro antioxidant activity of antioxidants and their ability to improve the oxidative stability of thermally processed meat products. Whether or not the antioxidant activity of 6-gingerol in meat cooking remains consistent with that in vitro needs to be experimentally demonstrated.

Therefore, LT muscle was selected in this research. Different concentrations of 6-gingerol were added during the heating process. Chemical methods, combined with intrinsic fluorescence spectra and FT-IR, were utilized to investigate the effects of 6-gingerol on the aggregation properties, oxidation degree, and molecular structure of the protein. Finally, SEM was employed to observe the protein microstructure and molecular docking was used to elucidate the mechanism. The aims were to investigate the relationship between 6-gingerol and the MP structure of beef and to provide a theoretical basis for finding the most suitable technology for beef storage and preservation, slowing down the oxidation rate, prolonging the storage period, and improving the economic efficiency and food quality of beef.

## 2. Materials and Methods

### 2.1. Materials and Chemical

LT muscle samples from cattle were sourced from Muye Natural Halal Fresh Beef and Mutton Shop in Luoyang City, Henan Province, in China. Twelve-month-old Simmental crossed local yellow cattle (male, castrated, weighing approximately 300–340 kg) were slaughtered according to commercial slaughter procedures. The sample was taken within 30 min after slaughter, transported to the laboratory within one hour of collection, and kept in cold storage (−18 ± 0.5 °C) for further use. 6-gingerol (purity ≥ 98%, CAS number: 23513-14-6) was obtained from Chengdu Desite Biotechnology Co., Ltd. (Chengdu, China). Bovine Serum Albumin (BSA) was purchased from Shanghai Blue Season Technology Development Co., Ltd. (Shanghai, China). All of the chemicals and reagents used in the study were of analytical grade and were purchased from Tianjin Deen Chemical Reagent Co., Ltd. (Tianjin, China).

### 2.2. Sample Preparation

After eliminating surface fat and connective tissue, the sample was minced into mincemeat with a meat grinder (Sinyder, 202, equipped with a 3 mm diameter plate.; Luoyang, China) and evenly divided into uniform pieces of approximately 5.0 ± 0.1 g per piece. The samples were randomly divided into five batches, and each batch was divided into the following two treatment groups: control and 40 μg/mL of 6-gingerol. The samples were heated at different temperatures (25, 50, 75, and 100 °C) for 6 min to simulate the cooking process of beef. The working concentration of 6-gingerol, temperature gradient, and heating time were selected according to the results of the preliminary experiment. It was found that the change in the effect of time on the beef samples showed a linear trend, while the effect of temperature showed a stepwise increase, so the subsequent experiments in this study were established on the premise of temperature change.

### 2.3. Extraction of Myofibrillar Protein

Myofibril isolations were carried out as described previously, with some modifications [18]. In total, 5.0 g of each sample was minced and homogenized (10,000 rpm) in four volumes of ice-cold buffer (0.1 mol/L of NaCl, 10 mmol/L of Na_3_PO_4_, 2 mmol/L of MgCl_2_, 1 mmol/L of EGTA, pH 7.0, 10 mmol/L of Na_2_HPO_4_, and 10 mmol/L of NaH_2_PO_4_) using a blender (60 s, 25 °C). The resulting homogenate was centrifuged at 10,000× *g* (10 min, 4 °C) and the supernatant was discarded. The sediment was then collected, re-suspended in the same buffer, and extracted again. After three repeated cycles of homogenization and centrifugation, the resulting sediment was added to four volumes of the other ice-cold buffer (0.1 mol/L of NaCl). The mixture was then homogenized and centrifuged at 5000× *g* (10 min, 4 °C). The collected precipitate was regarded as MP, which was stored at 4 °C and used within 48 h. Select MP samples were freeze-dried for subsequent FT-IR and SEM analyses. The obtained MP samples were dissolved in buffer (0.1 mol/L of Na_2_HPO_4_, 0.1 mol/L of NaH_2_PO_4_, and pH 7.0). The specific steps are shown in Figure 1.

### 2.4. Determination of Surface Hydrophobicity

The determination of surface hydrophobicity was conducted following a previous study by Cholh et al. [19] and slightly modified. The sample and bromophenol blue (BPB) were mixed at a ratio of 5/1 (*v*/*v*) and centrifuged at 5000× *g* (10 min, 4 °C). Then, 100 μL of supernatant was taken, to which 1 mL of water was added and mixed well, measured at 595 nm, and recorded as *A*_1_. In the blank group, the protein solution was diluted with 1 mL of phosphate buffer (pH 7.0) instead of the protein solution, measured at 595 nm, and recorded as *A*_0_. The surface hydrophobicity calculation formula is as follows:$$\mathrm{Surface}\;\mathrm{hydrophobicity}\left(\mu g\right)=200\times \frac{A_{1}-A_{0}}{A_{0}}$$

### 2.5. Determination of Total Sulfhydryl Content

Total sulfhydryl content was measured according to Yong et al. [20] and slightly modified. We took 1 mL of MP solution at a concentration of 5 mg/mL, added 9 mL of phosphate buffer and mixed well, took out 4 mL of the mixture and added 0.4 mL of 0.1% 5,5′-dithiobis (2-nitrobenzoic acid) DTNB solution, shook well, and then reacted in a water bath (25 min, 40 °C), with determination performed at 412 nm. The molar extinction coefficient of 13,600 mol/L^−1^ cm^−1^ was used to calculate the total sulfhydryl content.

### 2.6. Determination of Carbonyl Content

Carbonyl content was measured according to Fu et al. [21] and slightly modified. We took 1 mL of MP solution with a concentration of 5 mg/mL, added 1 mL of 10% trichloroacetic acid (TCA), shook well and centrifuged at 5000× *g* (5 min, 4 °C), discarded the supernatant, then added 1 mL of 2,4-dinitrophenylhydrazine (DNPH, dissolved in 2 mol/L of HCl) to the precipitate, then allowed it to react for 1 h at room temperature with 2 mol/L of HCl solution as a blank control group, then added 0.5 mL of 20% TCA, shook well, and centrifuged at 5000× *g* (4 °C). The protein precipitate was washed three times with ethanol-ethyl acetate solution (1:1 = *v*:*v*). In total, 1 mL from 6 mol/L of guanidine hydrochloride solution was added to the protein precipitate, which was shaken well and centrifuged at 5000× *g* (5 min, 4 °C) after reacting in a water bath (30 min, 37 °C). Absorbance was measured at 370 nm. The carbonyl content was calculated using a molar extinction coefficient of 22,000 mol/L^−1^ cm^−1^.

### 2.7. Determination of Protein Solubility

Protein solubility was measured according to Joo et al. [22] and slightly modified. The sample of MP solution was adjusted to 5 mg/mL with buffer solution, and 5 mL was taken and centrifuged at 5000× *g* (10 min, 4 °C). The concentration of MP before centrifugation was recorded as C0, and the concentration of MP in the supernatant after centrifugation was recorded as C1. The formula for the calculation of the solubility is as follows:$$\mathrm{Protein}\;\mathrm{solubility}\left(\%\right)=\frac{C1}{C0}\times 100$$

### 2.8. Determination of Zeta Potential and Particle Size

The Zetasizer Nano (BeNano 90, Dandong Baxter Instruments Co., Ltd., Dandong, China) was used to measure the Zeta potential and particle size [23]. The sample concentration was diluted to 1 mg/mL with distilled water and put into the measurement cell, and the following parameters were set: equilibrium time of 60 s, temperature of 25 °C, scattering angle of 90°, refractive index of the particles of 1.414, and absorption of 0.001.

### 2.9. Determination of Intermolecular Force

Intermolecular force was determined according to the method of Liu et al. [24], with a few modifications. The protein content in each solution was determined by the Caumas Blue method. A total of 2.000 ± 0.001 g of crushed beef was taken and mixed with 10 mL of solution A (0.05 mol/L of NaCl), solution B (0.6 mol/L of NaCl), solution C (0.6 mol/L of NaCl + 1.5 mol/L of urea), solution D (0.6 mol/L of NaCl + 8 mol/L of urea), and solution E (0.6 mol/L of NaCl + 8 mol/L of urea+0.5 mol/L of β-mercaptoethanol). The mixture was dispersed and homogenized at high speed for 2 min in an ice-water bath at 10,000 r/min and centrifuged at 10,000× *g* (15 min, 4 °C). The contribution of ionic bonding was the difference in the protein concentration between liquid A and liquid B. The contribution of hydrogen bonding was the difference between the liquid B and liquid C protein concentrations. The contribution of hydrophobic interactions was the difference between the liquid C and liquid D protein concentrations. The contribution of disulfide bonding was the difference between the liquid D and liquid E protein concentrations.

### 2.10. Determination of Intrinsic Fluorescence Spectra

Intrinsic fluorescence spectra were quantified following the method of [25], with slight modifications. The intrinsic fluorescence spectra of the diluted samples (1 mg/mL) at 300–400 nm (slit width 5 nm) were measured using a fluorescence spectrometer (Cary eclipse, Agilent, Santa Clara, CA, USA) at an excitation wavelength of 295 nm.

### 2.11. Determination of FT-IR Spectra

FT-IR analysis was performed according to a previous study [26]. Sample powder (1 mg) and KBr (100 mg) were mixed, compressed, and scanned by Brucker Fourier Transform Medium and a far-infrared spectrometer (Vector 33, BRUKER, Ettlingen, Germany). The test conditions were as follows: a scanning range of 400–4000 cm^−1^, a resolution of 4 cm^−1^, and 64 accumulations. After fitting with the second derivative and integral calculation using Peakfit 4.1.2 software, the infrared spectra were used to determine the relative contents of different secondary structures in MP.

### 2.12. Microstructure Observation

The protein microstructure was determined according to Tang et al. [27], with a few modifications. A scanning electron microscope (TM3030-Plus, Hitachi High-Tech Co., Ltd., Tokyo, Japan) was employed as the imaging instrument. MP samples were positioned at the base of a Petri dish, frozen at −20 °C for 5 h, and freeze-dried for 24 h. Following sputtering and gold plating, the surface of the samples was imaged using the scanning electron microscope, and their morphology was observed by varying the magnification levels.

### 2.13. Molecular Docking Simulation

Myosin accounts for 47.83% of the protein content of MP and is the main receptor for binding small molecules [28]. Therefore, myosin and 6-gingerol were selected for molecular docking. The amino acid sequence (accession number Q9BE40, consisting of 1938 amino acid residues) of bovine myosin was obtained from the UniProt Protein Data Bank (https://www.uniprot.org/, accessed on 10 January 2025), and the 3D structure of 6-gingerol was obtained from the PubChem database (https://pubchem.ncbi.nlm.nih.gov, accessed on 10 January 2025). Bovine myosin (receptor) and 6-gingerol (ligand) were imported into AutoDock Tools 4.2.6 software for hydrogenation, total charge calculation, and atom type assignment. Ligands and protein receptors were recorded in PDBQT format. Molecular docking simulations of bovine myosin and 6-gingerol were performed using the AutoDock 4.2.6 software to determine the optimal docking state based on the lowest docking energy. Three-dimensional structure visualization was expressed using the PyMOL molecular graphics system.

### 2.14. Statistical Analysis

Triplicate measurements were averaged before being statistically analyzed. Data were analyzed via one way analysis of variance (ANOVA) with SPSS (version 26.0, SPSS Statistics, IBM, New York, NY, USA), and the results are presented as mean ± standard deviation (X ± SD). All data were analyzed for the significance of differences (*p* < 0.05) by the means of Duncan’s new multiple range method. Graphs were created using Origin (version 2021, Origin Lab, Hampton, NY, USA).

## 3. Results and Discussion

### 3.1. Analysis of Surface Hydrophobicity of Beef MP During Cooking

BPB can bind to the hydrophobic amino acid residues of proteins, and the amount of binding reflects the surface hydrophobicity; the greater the amount of binding, the more hydrophobic the protein is [29]. Figure 2A shows the changes in surface hydrophobicity with an increasing temperature, with the surface hydrophobicity of all samples increasing significantly (*p* < 0.05). The explanation is that the heating and oxidation led to the unfolding of the MP structure, which increased the proportion of hydrophobic groups. It is remarkable that the increase in the surface hydrophobicity of the samples slowed down after treatment with 6-gingerol at 40 μg/mL. This shows that 6-gingerol was able to inhibit the large increase in the surface hydrophobicity of proteins under over-oxidation.

### 3.2. Analysis of Total Sulfhydryl Content of Beef MP During Cooking

Total sulfhydryl groups are the active sulfhydryl groups exposed on the surfaces of proteins and the sulfhydryl groups present inside the molecule. Total sulfhydryl group content is one of the key indicators for characterizing the degree of protein oxidation. As shown in Figure 2B, heating treatment caused a significant decrease (*p* < 0.05) in the total sulfhydryl content of all the samples compared with the control group without heating, which indicated that the cross-linking of disulfide bonds occurred in the oxidized MP. The total sulfhydryl content of unoxidized MP was 82.55 nmol/mg, which decreased by 18.54 after oxidation at 100 °C. However, the total sulfhydryl content of the 6-gingerol group decreased by 11.06 nmol/mg, which is much less than that of control group. The possible reason for this may be that the free radicals on protein molecules lead to changes in their spatial structure, exposing the sulfhydryl groups encapsulated within the protein, which can be more easily oxidized to disulfide bonds [30]. It has also been indicated that a loss of sulfhydryl groups may also be due to chemical interactions between them and polyphenolic compounds, resulting in the formation of sulfhydryl-quinone adducts [31].

### 3.3. Analysis of Carbonyl Content of Beef MP During Cooking

The formation of carbonyl groups in proteins is a distinctive feature of protein oxidation, and changes in this content can directly reflect the degree of protein oxidation. As a result, carbonyl content can be used as a key indicator to evaluate the level of protein oxidation [32]. Figure 2C depicts the changes in the carbonyl content under different heating temperatures. The carbonyl content of unoxidized beef protein at 25 °C was 2.27 nmol/mg, and the carbonyl content increased to 6.14 nmol/mg after oxidation at 100 °C. This suggests that the MP in beef was accompanied by oxidative reactions during heating treatment, which induced the formation of protein carbonyls in the meat, leading to an elevation in its carbonyl content. However, the rise in the MP carbonyl content of beef in the 6-gingerol group was inhibited (*p* < 0.05), and the strong free radical scavenging potency exhibited by 6-gingerol effectively curbed a further rise in carbonyl content. Polyphenols protected proteins from hydroxyl radical damage by scavenging free radicals and chelating metal ions, thus preventing protein oxidation. This is similar to the findings of Huang et al. [33] that the addition of 0.5% of mulberry polyphenols significantly protected proteins from oxidative attack.

### 3.4. Analysis of Protein Solubility of Beef MP During Cooking

The solubility of proteins can indirectly reflect their surface hydrophobicity, and usually, the higher the surface hydrophobicity, the lower the solubility. As shown in Figure 2D, the solubility of MP in all groups exhibited a decreasing trend with an increasing temperature (*p* < 0.05), which was opposite to the trend of surface hydrophobicity. This was mainly due to the oxidative denaturation of proteins by heating, which led to aggregation and precipitation. However, 6-gingerol reduced fat and protein oxidation by scavenging oxidizing free radicals and prevented oxidation-induced cross-linking and aggregation between protein molecules, thereby increasing protein solubility. The solubility of the 6-gingerol group was significantly higher than that of the control group at 100 °C, so 40 μg/mL of 6-gingerol could, indeed, prevent lipid and protein oxidation. The above results clearly reveal that 6-gingerol exhibited excellent antioxidant properties at a concentration of 40 μg/mL.

### 3.5. Analysis of Zeta Potential and Particle Size of Beef MP During Cooking

Zeta potential expresses the effective charge on the surface of a protein and is an important indicator of MP stability, assessing the magnitude of electrostatic interaction forces between charged biopolymers and reflecting the dispersion and aggregation of proteins [34]. Particle size can also visualize the degree of protein aggregation and changes in spatial structure. In general, the larger the particle size of a protein, the greater its degree of aggregation [35]. During heating, when proteins were broken into smaller aggregates, new surfaces were created, leading to an increase in particle size. Figure 3 shows the effects of 6-gingerol on the average particle size and Zeta potential of proteins. As observed in Figure 3A, the Zeta potentials of MP were all negative under heating conditions, and their absolute values gradually decreased with an increasing temperature. The decrease in the 6-gingerol group was smaller than that in the control group, indicating that 6-gingerol significantly elevated the absolute value of Zeta, which also implies that the electrostatic repulsion of the homosexual charges was stronger at this concentration gradient, resulting in a more stable protein solution with smaller molecules formed or dispersed particles.

### 3.6. Analysis of Intermolecular Force of Beef MP During Cooking

Proteins are composed of amino acid residues folded into a unique structure, such as one or more specific spatial conformations driven by disulfide bonds, hydrogen bonds, ionic bonds, hydrophobic interactions, and van der Waals forces. Disulfide bonds are formed by the oxidation of the sulfhydryl groups of two cysteine residues within or between the peptide chains of a protein molecule [36]. Hydrogen bonding is an important force that maintains the secondary structures of proteins such as α-helix, β-folding, and irregular coiling. Ionic bonds are chemical bonds formed by electrostatic forces between positive and negative ions, and hydrophobic interactions are the main driving force when protein molecules fold.

As can be seen from Table 1, the four intermolecular forces in each group of samples changed differently as the heating temperature increased, with ionic and hydrogen bonding showing a significant decreasing trend (*p* < 0.05) and disulfide bonding and hydrophobic interactions showing a significant increasing trend (*p* < 0.05). The probable reason for this was that heating broke the equilibrium within the forces, and the ionic and hydrogen bonds were broken one after the other, with the content being minimized when the temperature was raised to 100 °C. The increase in disulfide bonds was inextricably linked to the total sulfhydryl group, and heating and oxidation led to the cross-linking of the disulfide bonds, which mirrored the change in the total sulfhydryl group in the previous section. In stark contrast are the results in Table 2. Specifically, the ionic bonding content was higher than that of the control group without 6-gingerol after 40 μg/mL of 6-gingerol treatment in all cases, as was the hydrogen bonding, which is in agreement with the results of the study on the effect of cactus polysaccharides on the structure of myosin gel proteins by You et al. [37]. The reason may be that the addition of 6-gingerol affected the unfolding and folding of beef MP molecules during oxidation, as well as the exposure of the sulfhydryl and hydrophobic groups in the MP molecules, which could have had a hindering effect on the formation of disulfide bonds.

### 3.7. Intrinsic Fluorescence Spectra Analysis

Generally, tryptophan (Trp) residues hide in their natural state in the hydrophobic interior of proteins. When the protein was denatured and the protein conformation was unfolded, the Trp residues hidden in the hydrophobic interior of the protein were exposed to the solvent to accelerate the burst phenomenon. Figure 4 demonstrates the changes in the intrinsic tryptophan fluorescence intensity of MP under the different temperature treatments.

The fluorescence intensity decreased significantly after heating, and the fluorescence intensity decreased by more than 70% when the temperature was increased to 75 °C. It is evident that heating and oxidation induced the protein to be partially or completely unfolded, exposing more tryptophan portions, which, in turn, caused a change in the tertiary structure of MP. Figure 4A displays that the λ-max distribution of the MP in the control group was around 348 nm, and after heating at 75 °C, the λ-max experienced a shift towards lower wavenumbers (blue shift) and changes in intensity. This indicates that the microenvironment in which the Trp residues were located saw an increased hydrophobicity and decreased polarity. Figure 4B shows that the endogenous fluorescence intensity of the 6-gingerol group was reduced, exhibiting the fluorescence burst phenomenon. However, the fluorescence intensity after oxidation at 100 °C was slightly higher than that of the control group and was accompanied by a redshift of λ-max. Because 6-gingerol is a powerful fluorescence quencher, it led to the quenching of the characteristic fluorescence spectral peaks of tryptophan residues, which, in turn, changed the microenvironment near the MP tryptophan residues and gradually shifted them from a nonpolar environment to a polar environment [38]. In summary, 6-gingerol led to the gradual exposure of Trp residues to the protein surface and a decrease in fluorescence intensity, which meant that 6-gingerol inhibited the rise in MP fluorescence intensity, thus contributing to maintaining the stability of the beef MP tertiary structure.

### 3.8. FT-IR Spectra Analysis

The amide A band (3400 cm^−1^), amide I band (1654 cm^−1^), amide II band (1541 cm^−1^), and amide III band (1220 cm^−1^) reflected the N-H stretching vibration, the C=O stretching vibration, the N-H bending vibration and C-N stretching vibration and the N-H bending vibration and C-H stretching vibration, respectively [39].

As shown in Figure 5A, the amide A band was slightly red-shifted and the amide I band was shifted to the direction of a high wave number with a prolongation of the heating time. The characteristic spectral band reflecting the α-helix near 1655 cm^−1^ began to shift to 1670 cm^−1^, indicating that the α-helix of beef MP might have gradually transformed to β-turn and the secondary structure was changed. This was because heating broke the hydrogen bond between the carbonyl group and the amino group, leading to the splitting of the α-helix. Figure 5B demonstrates that the amide A band of beef MP shifted from 3292 cm^−1^ to 3321 cm^−1^ after the addition of 6-gingerol, which could be attributed to the N-H stretching vibration of the amide A band and the formation of hydrogen-bonding contacts, which confirms that the interaction between MP and 6-gingerol resulted in a change in the intensity of the characteristic spectral bands and the shifting of the absorption bands. The shape of the absorption peak of the heated protein amide I band was similar to that of the control, but the two wave number positions were different. Interestingly, after the addition of 6-gingerol, the total absorbance of the amino acid residues of MP was changed and the peak shapes were altered to different degrees. A peak at 2351 cm^−1^ appeared which was not previously present, i.e., 6-gingerol may have formed new covalent bonds with beef MP. In addition, the characteristic spectral peak at 1562 cm^−1^ appeared as the amide II band, where all bands were shifted to a low wave number direction. The reason for this shift was the bending vibration of N-H and the stretching vibration of C-N. Near 1232 cm^−1^, the peak shape gradually broadened until disappearing with an increase in 6-gingerol concentration, and the spectral bands were not significantly shifted [40]. The above results indicate that 6-gingerol changed the spectral shape of beef MP, shifted the spectral bands representing different regions to different degrees, and then changed the protein secondary structure.

### 3.9. Secondary Structure Analysis

The amide I region (1600–1700 cm^−1^) was analyzed in order to gain a deeper understanding of the changes in the secondary structure of MP after the addition of 6-gingerol. As shown in Figure 5C, the α-helix of beef MP decreased most significantly after heating at 100 °C, the β-folding and random curling increased, and the β-turn angle did not change significantly. Thus, beef MP was a typical secondary structure dominated by α-helix. Heating oxidation led to the unfolding of the protein molecules, which promoted the weakening of the hydrogen bonding interactions that maintain the stability of the α-helix. An explanation for this may be the interaction of the phenolics in 6-gingerol with the hydrogen bonds formed between carbonyl oxygen (C=O) and amino hydrogens (N-H), which maintained the stabilization of the α-helix, thereby disrupting the hydrogen bonding that maintained the α-helical structure and ultimately the unfolding of the protein [41]. The slower rate of decrease in the relative content of α-helix in the 6-gingerol group compared to the control group may be attributed to the fact that the phenolic hydroxyl group of 6-gingerol can effectively terminate the oxidative chain reaction by dehydrogenation to generate semiquinone radicals and, subsequently, biphenol. Figure 5D draws the conclusion that the α-helical structure of beef MP may be transformed into a β-folded and irregularly coiled structure, while the content of β-fold and irregularly coiled structures had a positive correlation with the water retention of the meat, as well as being the basis for protein aggregation and stabilization. Therefore, 6-gingerol facilitated the folding and orderly arrangement of the protein molecular structure, as well as the enhancement of electrostatic interactions between amino acids, which, in turn, maintained the stability of the secondary structure of beef MP.

### 3.10. SEM Analysis

SEM can be used to observe changes in tissue microstructure, and the effect of 6-gingerol on the microstructure of beef is depicted in Figure 6. A series of ordered morphological transformations were observed in beef MP magnified by electron microscopy at a magnification of 500×. Figure 6A presents the irregular and disordered network structure. With an increasing temperature, oxidized beef MP demonstrated a more open organizational architecture with significant gaps in its three-dimensional mesh structure, and it was hypothesized that this phenomenon originated from fatty acid depletion triggered by lipid peroxidation and protein cross-linking breaks, which led to the reorganization of the original ultrastructure of the myofibrils [42]. In contrast, Figure 6B shows that the beef MP of the 6-gingerol group exhibited a more uniform and tightly packed internal structure with fewer holes. Heating to 100 °C led to the highest degree of MP oxidation, a more pronounced antioxidant effect of 6-gingerol, the tightness of the structure tending to increase, the number of pores further contracting, and the spacing of myofibers also being reduced [43]. This series of changes was likely due to the inherent free radical scavenging ability of 6-gingerol, which effectively blocked the oxidative chain reaction in beef tissues, thus safeguarding the stability and integrity of beef MP. It can be inferred that 6-gingerol played a key role in optimizing the structural arrangement of beef myofibrillar proteins and enhancing the stability of their spatial organization.

### 3.11. Molecular Docking Analysis

Molecular docking is a powerful tool to visualize the interactions between receptors and ligands, such as proteins and polyphenols [44]. We used a protein homology model, as described by Wei et al. [45], which showed more than 90% agreement and was considered as suitable for molecular docking. In general, the binding energy for molecular docking is less than 0 kcal/mol, the molecules have binding interactions, and this binding is able to occur without the influence of external forces [46]. The molecular docking results are illustrated in Figure 7, which presents the types of valence bonds and key amino acid residues of the best docking positions between the protein receptor and the 6-gingerol.

As shown in Figure 7A, 6-gingerol was surrounded by amino acid residues of bovine myosin and the lowest binding energy with myosin was −8.758 kcal/mol, which indicates that the docking results were effective. Specifically, the compound can form two hydrogen bonds with ASN-447 on the protein, one Pi-Alkyl with LYS-437, one Alkyl with LEU-623, one Pi-Sigma with VAL-443, and one C-H bond with ALA-444. The hydrogen bonding interactions between 6-gingerol and myosin molecules were short and strong, with an average bond length of 4.86 Å. These interactions played a key role in slowing down the changes in the molecular structure properties of the MP molecules and maintaining the structural stability of proteins, and could greatly affect their biological functions. Additionally, 6-gingerol formed van der Waals forces with TRP-441, LEU-622, LEU-440, THR-468, ILE-266, and GLU-267 when docked to myosin, which further strengthened the non-covalent force between the two and resulted in a more robust binding (Figure 7B). Cao et al. [47] also showed that hydrogen bonding is one of the main forces in the interaction between phenolics and proteins, so the reduced hydrogen bonding content caused by oxidative denaturation may also lead to a decrease in the ability of myofibrillar proteins to bind to phenolics. Due to the multiple hydroxyl groups in the molecular structure of 6-gingerol, this provided a large number of hydrogen bonding sites for its interaction with bovine myosin, and the formation of hydrogen bonds contributed to the stability of the protein structure and reduced protein aggregation during the heating process. Figure 7C shows the changes in the hydrophobic binding sites between 6-gingerol and MP, and it can be seen that the interaction between the two was strongly hydrophobic, indicating that 6-gingerol had a strong adsorption capacity and was more likely to bind to MP through hydrophobic interactions.

## 4. Discussion

In this study, the antioxidant mechanism of 6-gingerol on beef MP was systematically investigated using multidisciplinary approaches. Specifically, the inhibitory effect of 6-gingerol on protein aggregation is hypothesized to arise from its simultaneous modulation of both covalent and noncovalent interactions. The phenolic hydroxyl group in 6-gingerol competitively scavenges free radicals that trigger disulfide bond rearrangements, thereby mitigating irreversible aggregation. Concurrently, its hydrophobic benzene ring disrupts the hydrophobic interactions between partially unfolded proteins, as corroborated by FT-IR evidence of retained α-helical structures. Moreover, the carbonyl moiety of 6-gingerol forms hydrogen bonds with lysine residues, counteracting the loss of ionic and hydrogen bonding during oxidation. However, several critical aspects remain underexplored. First, the dose-dependent effects of 6-gingerol on lipid-oxidation-derived flavor compounds (aldehydes and ketones) require systematic quantification. Second, the potential synergistic interactions between 6-gingerol and other natural antioxidants warrant further investigation. In future studies, the use of GC-MC to analyze the volatile compounds in meat products and to explore, in depth, the synergistic mechanism of 6-gingerol with other natural antioxidants, such as tea polyphenols, should become the focus of the research.

## 5. Conclusions

The results of this study showed that when the addition amount of 6-gingerol was 40 μg/mL, its antioxidant effect was the most significant. Fortunately, the application of 6-gingerol effectively inhibited the oxidation of fat and protein in beef during heating and hindered MP aggregation. Specifically, the total sulfhydryl reduction in the 6-gingerol group was 40.35% less than that in the control group and the carbonyl increase was 34.57% less than that in the control group. Furthermore, the protein conformation was altered, the intensity of intrinsic fluorescence was reduced, and aromatic amino acid residues were exposed to the protein surface. The secondary structure underwent rearrangement, N-H and C-N stretching vibrations and N-H bending vibrations were more pronounced, and the α-helix may have been transformed into a β-fold. SEM showed that the addition of 6-gingerol tended to make the organization of beef MP more compact, the size and number of pores decreased, and the spaces between muscle fibers narrowed accordingly. The mechanism of interaction between myosin and 6-gingerol was elucidated and the binding modes involved were mainly hydrogen bonds, van der Waals forces, and hydrophobic interactions. Therefore, the antioxidant activity of 6-gingerol in heat induction remained consistent with in vitro oxidation, and 40 μg/mL was its optimal action concentration. 6-gingerol should be considered as a natural antioxidant in the thermal processing of meat products to delay the oxidation process and maintain the stability of the MP structure. However, future research should be focused on the application of 6-gingerol to specific products, in which the effect of its application on the oxidative stability of meat products should be monitored.

## Figures and Tables

**Figure 1 foods-14-01081-f001:**
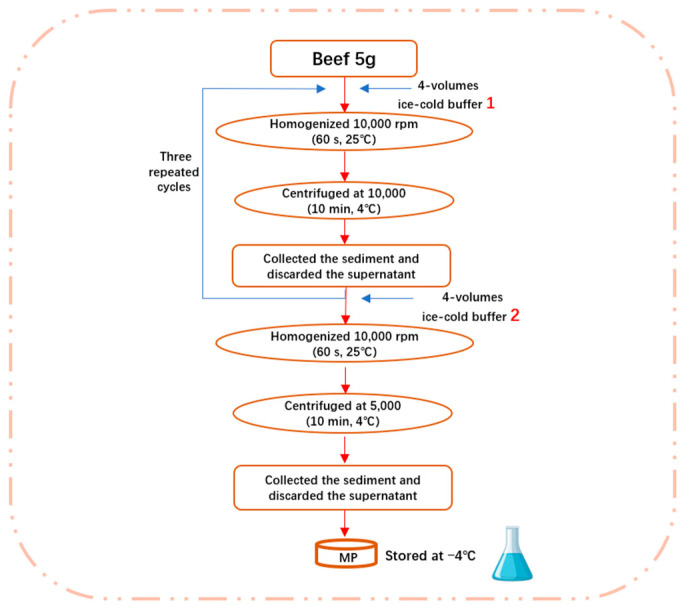
The main steps for extraction myofibrillar protein.

**Figure 2 foods-14-01081-f002:**
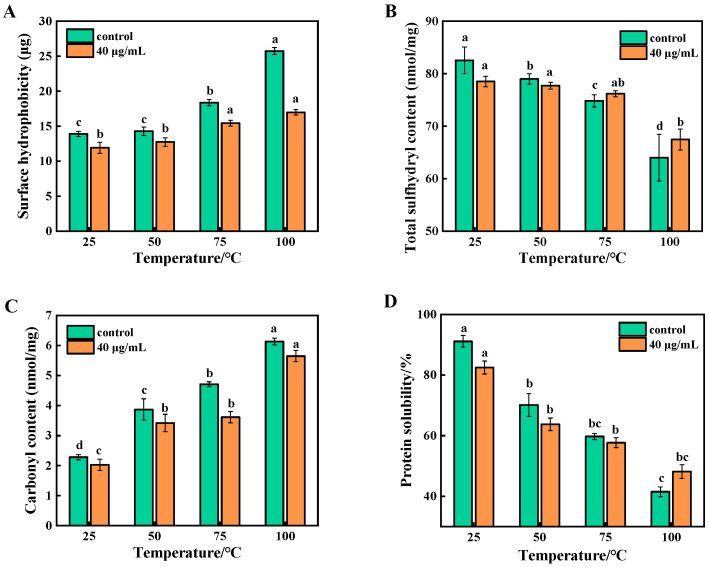
Changes in the surface hydrophobicity (**A**), total sulfhydryl content (**B**), carbonyl content (**C**), and protein solubility (**D**) of beef MP during heating. Note: letters (a–c) represent significant (*p* < 0.05) differences between different heating temperature in the same treatment group.

**Figure 3 foods-14-01081-f003:**
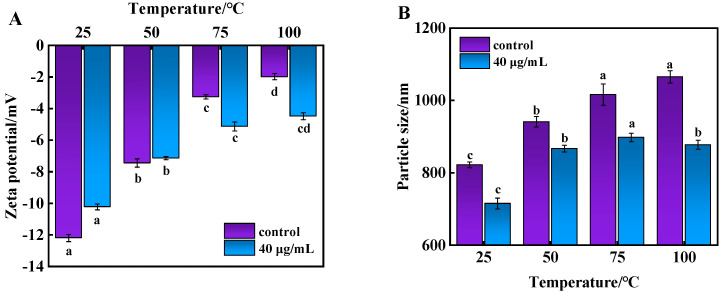
Changes in the Zeta potential (**A**) and particle size (**B**) of beef MP during heating. Note: letters (a–d) represent significant (*p* < 0.05) differences between different heating temperature in the same treatment group.

**Figure 4 foods-14-01081-f004:**
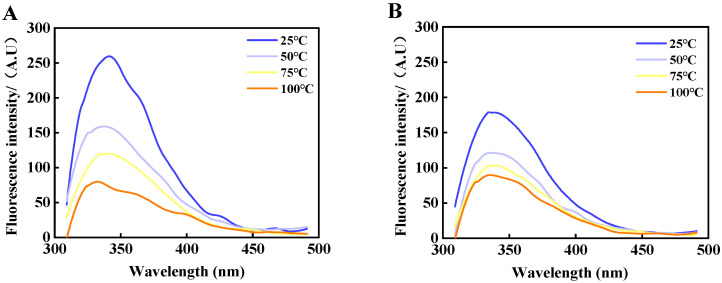
Intrinsic fluorescence spectra of beef MP during heating. Note: (**A**) represents control and (**B**) represents 6-gingerol concentration of 40 μg/mL.

**Figure 5 foods-14-01081-f005:**
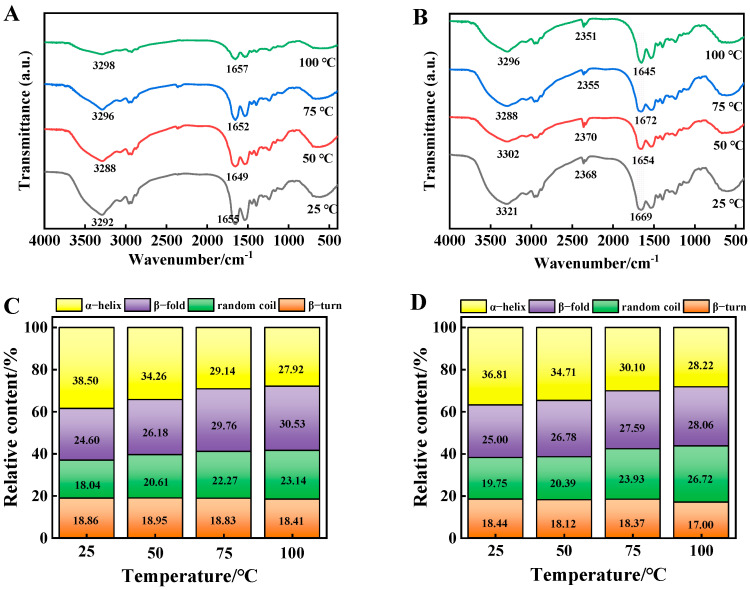
FT-IR spectra and secondary structure of MP during heating. Note: (**A**,**C**) represents control and (**B**,**D**) represents 6-gingerol concentration of 40 μg/mL.

**Figure 6 foods-14-01081-f006:**
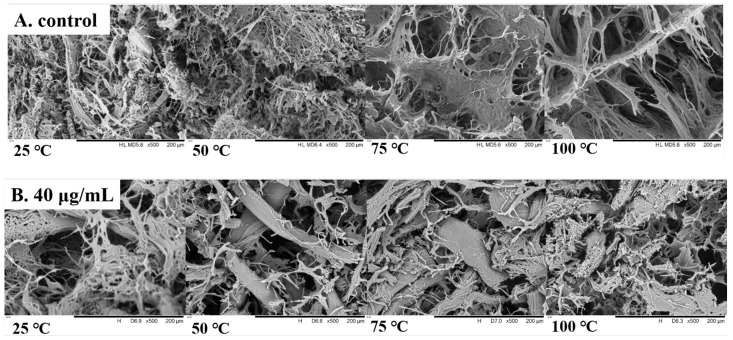
SEM photos microstructure of beef MP during heating.

**Figure 7 foods-14-01081-f007:**
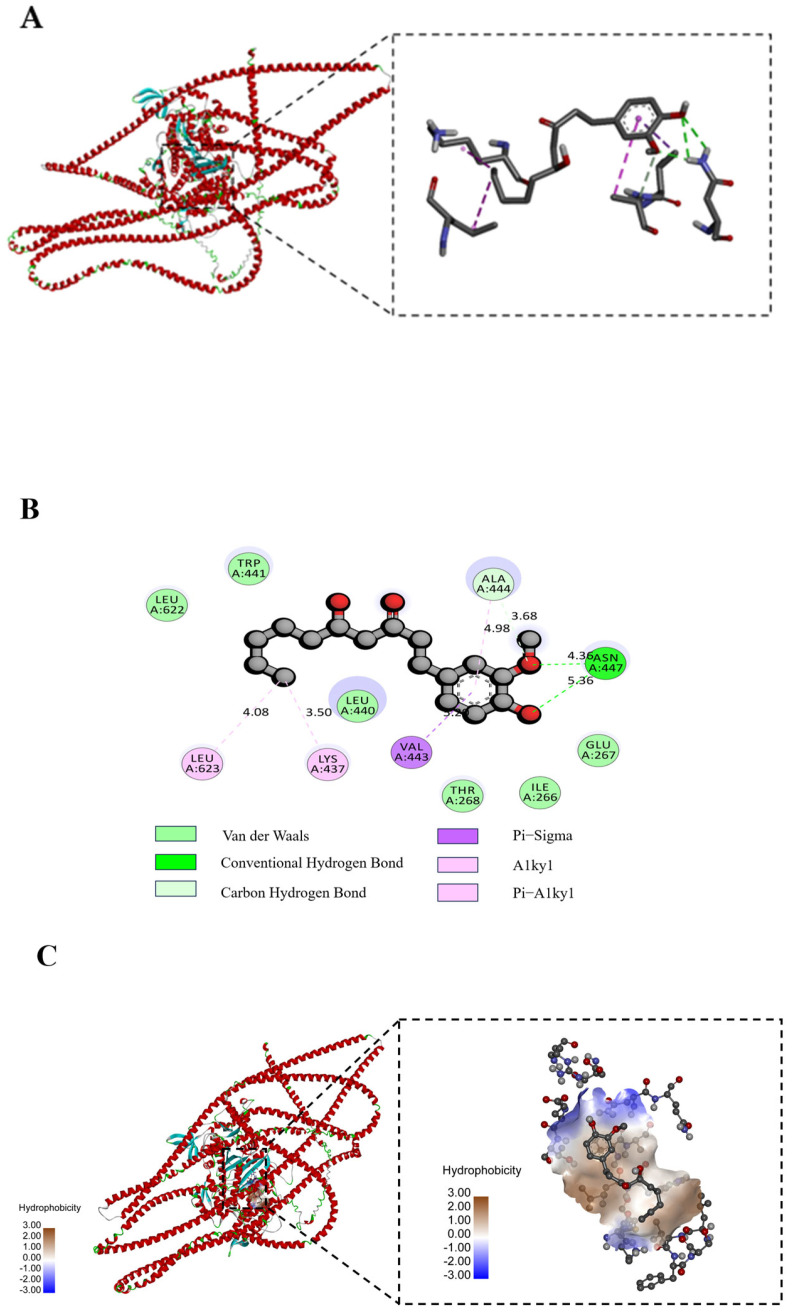
(**A**) Molecular docking conformation, (**B**) interactive 2D diagram, and (**C**) changes in hydrophobic binding sites bovine myosin and 6-gingerol.

**Table 1 foods-14-01081-t001:** Intermolecular force of beef MP during heating without 6-gingerol.

Temperature/°C	Ionic Bonds/%	Hydrogen Bonds/%	Disulfide Bonds/%	Hydrophobic Interactions/%
25	35.12 ± 0.23 ^a^	29.40 ± 0.77 ^a^	8.78 ± 0.37 ^d^	26.70 ± 0.90 ^b^
50	32.10 ± 0.39 ^b^	26.62 ± 0.34 ^b^	17.09 ± 0.51 ^c^	25.19 ± 0.17 ^b,c^
75	26.58 ± 0.38 ^c^	24.42 ± 0.73 ^c^	22.20 ± 0.77 ^b^	26.80 ± 0.47 ^b^
100	20.98 ± 0.16 ^d^	18.86 ± 0.42 ^d^	27.50 ± 0.90 ^a^	32.66 ± 0.76 ^a^

Note: Different lowercase letters indicate a significant difference (*p* < 0.05).

**Table 2 foods-14-01081-t002:** Intermolecular force of beef MP during heating with 40 μg/mL 6-gingerol.

Temperature/°C	Ionic Bonds/%	Hydrogen Bonds/%	Disulfide Bonds/%	Hydrophobic Interactions/%
25	36.61 ± 0.48 ^a^	28.58 ± 1.07 ^a^	9.46 ± 0.5 ^d^	25.35 ± 1.42 ^a,b^
50	34.01 ± 0.16 ^b^	28.16 ± 0.71 ^a,b^	11.72 ± 0.52 ^c^	26.11 ± 0.86 ^a^
75	33.80 ± 0.45 ^b^	25.38 ± 1.51 ^b^	15.92 ± 0.82 ^a,b^	24.90 ± 0.92 ^b^
100	31.57 ± 0.12 ^c^	25.19 ± 0.63 ^b^	16.88 ± 1.48 ^a^	26.36 ± 0.77 ^a^

Note: Different lowercase letters indicate a significant difference (*p* < 0.05).

## Data Availability

The original contributions presented in the study are included in the article, further inquiries can be directed to the corresponding author.

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
