# Peer review of "Effect of 6-Gingerol on Oxidation and Structure of Beef Myofibrillar Protein During Heating"

_foods, 2025, doi:10.3390/foods14071081_

Round 1

Reviewer 1 Report

Comments and Suggestions for Authors

C1.

Line 83 – 87

Please provide reference for the following statement „In recent years, there have been numerous studies employing natural antioxidants to mitigate the oxidation of meat products in thermal processing, such as tea polyphenols (with strong in vitro antioxidant activity) can improve protein oxidation and fat oxidation in grilled beef patties and grilled pork patties, thus improving their quality.

C2.

Line 95 – 97

Please check weather word „intrinsic“ should be capitalized in the following sentence „Chemical methods, combined with Intrinsic fluorescence spectra and FT-IR were utilized to investigate the effects of 6-gingerol on the aggregation properties, oxidation degree, and molecular structure of the protein.“

C3.

Line 107 – 109

Authors should check information regarding weight of slaughtered cattle. 30 – 34 kg seems low for 12 month old cattle. Maybe the authors wanted to write 300 – 340 kg. In addition, please provide information regarding the breed of cattle.

C4.

Line 117-118

I kindly ask the authors to provide information on the diameter of plate used for mining of meat.

C5.

Line 294

Resolution of Figure 2 should be higher. It is difficult for readers to read all the text inside this figure. Good example is Figure 3, which resolution is good and easily readable.

C6.

Line 490

In conclusion, authors should consider writing a sentence about downsides of the experiment. In my opinion, after application of antioxidants, products should be left in storage for several days where the effect of time on oxidation can be monitored.

In addition, authors should explain potential practical implementation of the use of this antioxidant and to give suggestions for further research (for example application in cooked meats, fermented sausages). Here we can raise a question of the potential effect of this antioxidant on sensory characteristics of processed meat in the first place in terms of taste and aroma.

C7.

Line 504 – 508

Since the authors used minced meat in the experiment, it is not proven whether or not application of 6-gingerol would have the same effect on products that are dried in one piece, like dry ham or dried M. longissimus dorsi. I believe that the following statement „.6-gingerol can be added as a natural antioxidant in thermal processing of meat products to retard the oxidation process and maintain the stability of MP structure.“ can be stated only after experiments on specific products.

Since this is not the case, your statement should be similar to the following „Therefore, the antioxidant activity of 6-gingerol in heat induction remained consistent with in vitro oxidation, and 40 μg/mL was its optimal action concentration. 6-gingerol should be considered as a natural antioxidant in thermal processing of meat products to retard the oxidation process and maintain the stability of MP structure. However, future research should be focused on application of 6-gingerol to specific products in which the effect of its application on oxidative stability of meat products should be monitored“.

Author Response

Comments 1: (Line 83 – 87) Please provide reference for the following statement „In recent years, there have been numerous studies employing natural antioxidants to mitigate the oxidation of meat products in thermal processing, such as tea polyphenols (with strong in vitro antioxidant activity) can improve protein oxidation and fat oxidation in grilled beef patties and grilled pork patties, thus improving their quality.

Response 1: Thank you very much for your valuable suggestion. I have added two references as required. Your comment has significantly improved the comprehensiveness of my work (Page 16, Lines 584-588).

Comments 2: (Line 95 – 97) Please check weather word „intrinsic“ should be capitalized in the following sentence „Chemical methods, combined with Intrinsic fluorescence spectra and FT-IR were utilized to investigate the effects of 6-gingerol on the aggregation properties, oxidation degree, and molecular structure of the protein.“

Response 2: Thank you for pointing out the inconsistency in capitalization. The word "intrinsic" has been revised to lowercase in the sentence as suggested. Your attention to detail has greatly improved the clarity of the manuscript (Page 3, Lines 98).

Comments 3: (Line 107 – 109) Authors should check information regarding weight of slaughtered cattle. 30 – 34 kg seems low for 12 month old cattle. Maybe the authors wanted to write 300 – 340 kg. In addition, please provide information regarding the breed of cattle.

Response 3: Thank you for your careful review and insightful feedback. We sincerely apologize for the inadvertent error regarding the cattle weight. The values have been corrected to "300–340 kg" in the revised manuscript, and we have now specified the cattle breed as " Simmental crossed local yellow cattle" in the relevant section. Your attention to detail has significantly enhanced the accuracy of our work (Page 3, Lines 110-111).

Comments 4: (Line 117 – 118) I kindly ask the authors to provide information on the diameter of plate used for mining of meat.

Response 4: Thank you for your valuable suggestion. As requested, we have added detailed information about the meat mincing equipment in the revised manuscript, including the brand (Sinyder), model (202), and diameter of the plate (3 mm). Your feedback has enhanced the methodological clarity of our work (Page 3, Lines 122-123).

Comments 5: (Line 294) Resolution of Figure 2 should be higher. It is difficult for readers to read all the text inside this figure. Good example is Figure 3, which resolution is good and easily readable.

Response 5: Thank you for your constructive feedback. We have revised Figure 2 to ensure its resolution matches that of Figure 3, and all Figures are now clearly visible. Your suggestion has significantly improved the readability of the figures (Page 7, Lines 298).

Comments 6: (Line 400) In conclusion, authors should consider writing a sentence about downsides of the experiment. In my opinion, after application of antioxidants, products should be left in storage for several days where the effect of time on oxidation can be monitored.

In addition, authors should explain potential practical implementation of the use of this antioxidant and to give suggestions for further research (for example application in cooked meats, fermented sausages). Here we can raise a question of the potential effect of this antioxidant on sensory characteristics of processed meat in the first place in terms of taste and aroma.

Response 6: We are deeply grateful for your insightful and constructive feedback, which has significantly elevated the quality of our manuscript. Your emphasis on temporal monitoring and sensory impacts has highlighted critical aspects that will guide our future research direction. While we fully recognize the effects of monitoring temporal changes on oxidation, the central scientific question of this study is the molecular regulation of the antioxidant mechanism of beef MP by 6-gingerol in response to a temperature gradient. Future studies could incorporate spectroscopic techniques to monitor the antioxidant effect of time on 6-gingerol in real time, which would provide new perspectives to reveal the temperature-time synergistic effect. Furthermore. we would like to express our deep appreciation and attach great importance to your suggestion on the potential effects of 6-gingerol on the flavor, aroma and organoleptic properties of meat products. Due to the length and focus constraints of the paper, we will elaborate on this section in a future paper. Thank you again for your insightful comments and guidance. 

Comments 7: (Line 504 – 508) Since the authors used minced meat in the experiment, it is not proven whether or not application of 6-gingerol would have the same effect on products that are dried in one piece, like dry ham or dried M. longissimus dorsi. I believe that the following statement „6-gingerol can be added as a natural antioxidant in thermal processing of meat products to retard the oxidation process and maintain the stability of MP structure.“ can be stated only after experiments on specific products.

Since this is not the case, your statement should be similar to the following „Therefore, the antioxidant activity of 6-gingerol in heat induction remained consistent with in vitro oxidation, and 40 μg/mL was its optimal action concentration. 6-gingerol should be considered as a natural antioxidant in thermal processing of meat products to retard the oxidation process and maintain the stability of MP structure. However, future research should be focused on application of 6-gingerol to specific products in which the effect of its application on oxidative stability of meat products should be monitored“.

Response 7: We are deeply grateful for your meticulous review and invaluable feedback, which have significantly contributed to the improvement of our manuscript. In the Conclusion section (Page 15, Lines 528-530), we have refined our statements to “6-gingerol should be considered as a natural antioxidant in thermal processing of meat products to retard the oxidation process and maintain the stability of MP structure. However, future research should be focused on application of 6-gingerol to specific products in which the effect of its application on oxidative stability of meat products should be monitored.”  Once again, we sincerely appreciate your dedication to advancing scientific rigor.

4. Response to Comments on the Quality of English Language

Point 1: The English is fine and does not require any improvement.

Response 1: Thank you for your kind feedback on the language quality of the manuscript. We are grateful to know that the English expression meets the required standards. We will ensure the clarity and accuracy of the text is maintained throughout the final version.

5. Additional clarifications

Your detailed feedback has been instrumental in improving our manuscript. All concerns have been addressed, and we are confident the revisions meet the journal’s requirements. Please let us know if further adjustments are needed. Thank you for your time and expertise—your contributions are deeply appreciated.

Reviewer 2 Report

Comments and Suggestions for Authors

Title: Effect of 6-Gingerol on Oxidation and Structure of Beef Myofibrillar Protein During Heating

(Overall)

This study explores the effects of 6-gingerol, a natural phenolic compound derived from ginger, on the oxidation and structural stability of beef myofibrillar proteins (MP) during heating. The research is timely and relevant, given the increasing consumer demand for clean-label and natural additives in meat processing. The originality of the study lies in its focus on both oxidation inhibition and protein structure stabilization, areas less explored compared to lipid oxidation studies. The comprehensive approach, utilizing multiple analytical techniques such as chemical assays, fluorescence spectroscopy, FT-IR, SEM, and molecular docking, strengthens the reliability of the findings. Identifying the optimal concentration of 6-gingerol at 40 μg/mL further adds to the practical significance of the study. However, the manuscript could benefit from a broader comparative discussion involving other natural antioxidants like rosemary extract and green tea polyphenols to highlight the relative advantages of 6-gingerol.

(Areas for improvement)

  1. The study effectively demonstrates the antioxidant capabilities of 6-gingerol but falls short in exploring its potential sensory impacts, such as flavor, aroma, and texture in meat products. Including sensory evaluations and functional tests, such as texture profile analysis and GC-MS analysis for volatile compounds, would provide a more comprehensive understanding of the practical implications of using 6-gingerol in meat processing. Additionally, a comparative analysis with synthetic antioxidants like BHA and BHT could strengthen the argument for replacing synthetic additives with natural ones. Cost-effectiveness and regulatory considerations for the commercial use of 6-gingerol also deserve more detailed discussion.
  2. While the study presents detailed data on the inhibition of protein oxidation through sulfhydryl and carbonyl content analysis, the explanation of how 6-gingerol prevents protein aggregation could be expanded. A deeper analysis of intermolecular interactions, including ionic bonds and disulfide bridges, might offer further insights into the mechanisms behind the observed stabilization effects. The preservation of the α-helical structure and the prevention of β-sheet formation are promising findings, suggesting that 6-gingerol could help maintain water-holding capacity and improve the juiciness of meat products. However, validating these results with additional methods, such as NMR spectroscopy, could enhance the credibility of the conclusions.
  3. The molecular docking results highlight the significance of hydrogen bonds and hydrophobic interactions between 6-gingerol and myosin, with a binding energy of -8.758 kcal/mol indicating strong and stable interactions. This aspect of the study is particularly commendable as it bridges the gap between theoretical and practical applications of natural antioxidants. Nevertheless, future studies should consider exploring the synergistic effects of 6-gingerol with other natural antioxidants to potentially enhance stability and extend shelf-life further.

Author Response

Comments 1: The study effectively demonstrates the antioxidant capabilities of 6-gingerol but falls short in exploring its potential sensory impacts, such as flavor, aroma, and texture in meat products. Including sensory evaluations and functional tests, such as texture profile analysis and GC-MS analysis for volatile compounds, would provide a more comprehensive understanding of the practical implications of using 6-gingerol in meat processing. Additionally, a comparative analysis with synthetic antioxidants like BHA and BHT could strengthen the argument for replacing synthetic additives with natural ones. Cost-effectiveness and regulatory considerations for the commercial use of 6-gingerol also deserve more detailed discussion.

Response 1: We sincerely appreciate your thoughtful questions regarding the sensory and functional impacts of 6-gingerol. We fully acknowledge the critical importance of texture analysis, aroma profiling, and comparative studies with synthetic antioxidants for practical implementation. In fact, sensory evaluation experiments and texture analysis have been completed by the team. Due to the journal's requirement of focusing on the core innovation points and the limitation of the journal's space constraints, they were not elaborated in the paper. We have strategically placed a core focus on antioxidant mechanisms. Previous experimental data showed that 6-gingerol did not have a significant effect, neither negative nor positive, on the sensory evaluation of protein foods. However, it is encouraging to note that your suggested GC-MS analysis of volatile compounds points us in the direction of a breakthrough! We plan to systematically analyze the antioxidant effect of 6-gingerol on lipids and its effect on key flavor products (aldehydes and ketones) of lipids in subsequent studies, and to construct a two-dimensional model of “antioxidant-flavor modulation” by combining electronic nose and sensory genomics techniques. We are very inspired by your insights on flavor, aroma and texture analysis, and even realize that this direction is sufficient to form an excellent paper to explore the sensory modulation mechanism of natural antioxidants in depth! In addition, we have included in the introduction a comparison of the antioxidant capacity of 6-gingerol with that of BHA/BHT, emphasizing the key advantages of its non-toxicity, wide availability, low extraction cost and extraction process maturity (Page 2, Lines 64-67 and 75-78). Thanks again for your suggestions.

Comments 2: While the study presents detailed data on the inhibition of protein oxidation through sulfhydryl and carbonyl content analysis, the explanation of how 6-gingerol prevents protein aggregation could be expanded. A deeper analysis of intermolecular interactions, including ionic bonds and disulfide bridges, might offer further insights into the mechanisms behind the observed stabilization effects. The preservation of the α-helical structure and the prevention of β-sheet formation are promising findings, suggesting that 6-gingerol could help maintain water-holding capacity and improve the juiciness of meat products. However, validating these results with additional methods, such as NMR spectroscopy, could enhance the credibility of the conclusions.

Response 2: We sincerely thank you for your invaluable suggestions, which have significantly deepened our mechanistic understanding of the research findings. In response to your recommendation, we have added a new paragraph (Page 14, Lines 495-503) in the discussion that integrates sulfhydryl/carbonyl and secondary structure analysis to systematically illustrate how 6-gingerol inhibits protein aggregation through regulation of disulfide bonds, hydrophobic interactions, and hydrogen bonding. Regarding the NMR technique, our team has already investigated the water migration pattern of beef at different temperatures using the LF-NMR system, which was published in the Chinese journal Food and Fermentation Industry in 2023 ([1]). In this study, we have augmented our results with complementary evidence from Fourier Transform Infrared (FT-IR) spectroscopy and molecular docking simulations, which collectively provide robust support for the proposed mechanisms. These methods allowed us to visualize conformational changes at both macro- and micro-scales, thereby maintaining the scientific integrity of our study.

[1]      WANG Ting-Min, XIE An-Guo, FU Shu-Ya, et al. Study on the effect of heating temperature on beef moisture based on low-field nuclear magnetic-near-infrared two-dimensional correlation spectroscopy. Food and Fermentation Industry,2023,49(20):288-293. DOI:10.13995/j.cnki.11-1802/ts.032552.

Comments 3: The molecular docking results highlight the significance of hydrogen bonds and hydrophobic interactions between 6-gingerol and myosin, with a binding energy of -8.758 kcal/mol indicating strong and stable interactions. This aspect of the study is particularly commendable as it bridges the gap between theoretical and practical applications of natural antioxidants. Nevertheless, future studies should consider exploring the synergistic effects of 6-gingerol with other natural antioxidants to potentially enhance stability and extend shelf-life further.

Response 3: Thank you for recognizing the importance of our molecular docking analysis and for your visionary suggestions regarding antioxidant synergism. We have fleshed out the manuscript and made an outlook at the end of the discussion. In future studies, the GC-MC analysis of 6-gingerol and its combination with tea polyphenols will be the direction of our exploration will be the direction of our team (Page 14, Lines 507-509).

4. Response to Comments on the Quality of English Language

Point 1: The English is fine and does not require any improvement.

Response 1: Thank you very much for your positive feedback regarding the English in our manuscript. We are glad that the language meets the required standard. This is also a result of our efforts to ensure clear and accurate communication of our research findings.

5. Additional clarifications

Thank you for your thorough review and guidance. We have carefully addressed all concerns raised and are confident the manuscript now meets the journal’s standards. Should you require any additional clarifications or adjustments, please do not hesitate to contact us. We deeply appreciate your time and expertise in advancing this work.

Round 2

Reviewer 1 Report

Comments and Suggestions for Authors

The manuscrpt can be accepted.

Reviewer 2 Report

Comments and Suggestions for Authors

Because the authors have revised this manuscript well, reflecting the opinions of reviewers, it is believed that this manuscript can be published in this journal.